# Microrna-486-5P Regulates Human Pulmonary Artery Smooth Muscle Cell Migration via Endothelin-1

**DOI:** 10.3390/ijms231810400

**Published:** 2022-09-08

**Authors:** Ting-An Yen, Hsin-Chung Huang, En-Ting Wu, Heng-Wen Chou, Hung-Chieh Chou, Chien-Yi Chen, Shu-Chien Huang, Yih-Sharng Chen, Frank Lu, Mei-Hwan Wu, Po-Nien Tsao, Ching-Chia Wang

**Affiliations:** 1Department of Pediatrics, National Taiwan University Children Hospital, National Taiwan University College of Medicine, Taipei 100, Taiwan; 2Department of Surgery, National Taiwan University Hospital, National Taiwan University College of Medicine, Taipei 100, Taiwan

**Keywords:** pulmonary arterial hypertension, miR-486-5p, pulmonary artery smooth muscle cells

## Abstract

Pulmonary arterial hypertension (PAH) is a fatal or life-threatening disorder characterized by elevated pulmonary arterial pressure and pulmonary vascular resistance. Abnormal vascular remodeling, including the proliferation and phenotypic modulation of pulmonary artery smooth muscle cells (PASMCs), represents the most critical pathological change during PAH development. Previous studies showed that miR-486 could reduce apoptosis in different cells; however, the role of miR-486 in PAH development or HPASMC proliferation and migration remains unclear. After 6 h of hypoxia treatment, miR-486-5p was significantly upregulated in HPASMCs. We found that miR-486-5p could upregulate the expression and secretion of ET-1. Furthermore, transfection with a miR-486-5p mimic could induce HPASMC proliferation and migration. We also found that miRNA-486-5p could downregulate the expression of SMAD2 and the phosphorylation of SMAD3. According to previous studies, the loss of SMAD3 may play an important role in miRNA-486-5p-induced HPASMC proliferation. Although the role of miRNA-486-5p in PAH in in vivo models still requires further investigation and confirmation, our findings show the potential roles and effects of miR-486-5p during PAH development.

## 1. Introduction

Pulmonary arterial hypertension (PAH) is a fatal or life-threatening disorder characterized by elevated pulmonary arterial pressure and pulmonary vascular resistance. Although there are many treatment strategies for PAH, the therapies tend to fail to target the underlying cellular and molecular abnormalities [1]. PAH is characterized by abnormal vascular remodeling, including the proliferation and phenotypic modulation of pulmonary artery smooth muscle cells (PASMCs) [2,3,4]. Several factors impact the abnormal proliferation of PASMCs, and hypoxia is a well-known stimulus for the development of PAH [5]. During hypoxia, hypoxia-inducible factor-1 (HIF-1), as a nuclear transcription factor, functions as a main regulator of the adjustment to hypoxia [6]. Experimental and clinical data suggest that HIF-1α activation exacerbates the progression of pulmonary hypertension.

Vasoactive peptide endothelin-1 (ET-1), induced by HIF-1 during hypoxia, plays a critical role in abnormal PASMC growth [7]. ET-1, which is secreted by endothelial cells and smooth muscle cells, binds with the ET-1 receptor (ETA and ETB) on PASMCs and activates its downstream signaling such as the MAPK pathway, which leads to PASMC proliferation, inflammation, and migration [8,9,10]. Several studies demonstrated treatment with an ET-1-receptor antagonist to be a therapy for hypoxic pulmonary hypertension in animal models [9,10] and for effective treatment in patients [11]. Furthermore, a recent study focused on the roles of miRNAs in the regulation of ET-1 expression and secretion, and tried to establish a new strategy for PAH treatment [12].

MicroRNAs (miRNAs) are a group of endogenous noncoding RNA molecules that are 21–24 nucleotides in length, and negatively regulate gene expression at the post-transcriptional level through targeting specific messenger RNAs (mRNAs) and promoting mRNA degradation [13]. The aberrant miRNA expression that enhances the proliferation and reduces the apoptosis of pulmonary artery smooth muscle cells is a key characteristic in the pathogenesis of PAH [14,15]. miRNA-124 is downregulated by hypoxia in HPASMCs, and the overexpression of miRNA-124 suppresses the NFAT pathway and inhibits the proliferation of HPASMCs [16]. miRNA-338 could modulate pulmonary-hypertension-associated endothelial dysfunction with implications for pulmonary vascular and respiratory function through directly binding to the 3′-UTR of the HIF-1α mRNA, which causes abnormal ET-1 expression [17]. Similarly, the expression of miRNA-17 is increased in HPASMCs under hypoxia and in the lung tissues of PAH patients, and miRNA-17 could regulate the proliferation and apoptosis of HPASMCs through mitofusin 2 [18]. Those studies indicated that miRNAs are essential for PAH development. However, there are many different types of unknown miRNAs that may also be involved in PAH progression. It is necessary to conduct more studies to clarify the mechanism of miRNAs in PAH.

PASMC proliferation and migration are among the most critical pathological changes during PAH. Exposure to hypoxia upregulates miR-486 expression in bone-marrow-derived mesenchymal stem cells (BM-MSCs) [19]. miR-486 functions as a hypoxia-response miRNA by mediating the hypoxia-induced proliferation and survival of BM-MSCs through regulating AKT signaling. However, the role of miR-486 in PASMC proliferation and migration remains unclear. In this study, we attempt to demonstrate that the dysregulation of miR-486-5p in HPASMCs would be a potential factor promoting PAH development.

## 2. Results

### 2.1. Alteration of miRNA Expression under Hypoxia Treatment in Isolated HPASMCs

To establish a representative in vitro model for PAH progression, we collected tissues from main or peripheral pulmonary arteries and isolated HPASMCs from a 6-month 10-day-old patient with a diagnosis of the tetralogy of Fallot (TOF) which is a common congenital heart disease (CHD) in children. TOF was diagnosed through Level II fetal screening and the patient received surgical repair at the age of six months. According to the previous literature, 4–15% of patients with CHD develop PAH. The prevalence of postoperative PAH-associated TOF is 1%. Compared with normal HPASMCs or other cell lines, HPASMCs isolated from CHD patients are more suitable for evaluating the mechanism of PAH development. 

To investigate the possible mechanisms involved in PAH development, we evaluated miRNA expression in HPASMCs under 6 h of hypoxia exposure through next-generation sequencing (NGS). In this study, NGS is a high-throughput screening tool for evaluating miRNA changes. We did not repeat NGS multiple times. However, we used real-time PCR assay to confirm the results of NGS. Table 1 shows the miRNAs whose expression increased or decreased more than twofold after hypoxia treatment. The expression of miR-486-5p was markedly induced (2.8-fold) after hypoxia exposure. The levels of miR-486-5p detected by qRT-PCR showed a pattern similar to that shown by the NGS results. As shown in Figure 1A, the expression of miRNA-486-5p detected by qRT-PCR was significantly increased after 6, 16, and 24 h of hypoxia exposure. miRNA-486 plays important roles in the growth and migration of different cancer cells, which are also critical mechanisms for inducing PAH development [20,21,22]. However, the role of miR-486-5p in PAH remains unclear. On the basis of the results obtained from NGS and RT-PCR, miR-486-5p was chosen to be our major target to investigate its possible role in PAH. 

### 2.2. miR-486-5p Induced HPSMC Proliferation and Migration Ability

The proliferation and migration of smooth muscle cells are important factors for vascular-wall remodeling in PAH development. ET-1 release from endothelial cells and smooth muscle cells is one of the key factors promoting the proliferation and inhibiting the apoptosis of smooth muscle cells [23]. We also tried to evaluate the effect of miR-486-5p on ET-1 expression and secretion. The transfection of a miR-486-5p mimic induced ET-1 mRNA and protein expression (Figure 1B,D). miR-486-5p could also increase the secretion of ET-1 into the medium (Figure 1C). Furthermore, HPSMC proliferation was significantly increased after miR-486-5p transfection (Figure 2A). In a wound-healing assay, HPASMCs transfected with the miR-486-5p mimic also showed greater migration ability compared with that of the control group (Figure 2B). On the basis of our data, miR-486-5p could increase the expression and secretion of ET-1, and may induce HPASMC proliferation and migration through the autocrine effect of ET-1.

### 2.3. miR-486-5p Regulated Smad2 and Smad3 in HPSMCs Isolated from a CHD Patient

Previous studies showed that Smad2 is one of miR-486-5p’s targets [24,25,26]. Furthermore, the TGFβ-1/Smad pathway is an important pathway regulating PAH development [27,28,29]. We investigated whether miR-486-5p could regulate the expression and phosphorylation of Smad2 and Smad3 in isolated HPASMCs during PAH development. The transfection of the miR-486-5p mimic could downregulate the expression of Smad2 at the mRNA and protein levels. However, the phosphorylation of Smad2 was not affected after miR-486-5p overexpression (Figure 3). The expression of Smad3 was not affected by miR-486-5p transfection. However, the phosphorylation of Smad3 was decreased in a dose-dependent manner (Figure 3A,B). On the basis of our data, miR-486-5p had the ability to inhibit the Smad 2/3 pathway in HPASMCs isolated from a CHD patient, and may regulate PAH development through this pathway.

## 3. Discussion 

In this study, the level of miR-486-5p increased significantly in HPASMCs isolated from a CHD patient in an in vitro PAH model. We first found that miR-486-5p could regulate the expression and secretion of ET-1, which is a critical factor causing HPASMC proliferation and migration during PAH development. Our data show that miR-486-5p may play an important role in the remodeling of the lung vascular wall. Furthermore, miR-486-5p could regulate the activity of Smad2/3, which is also an important signaling pathway involved in PAH-induced vascular remodeling. 

Diverse roles of miR-486-5p in cell proliferation and migration have been demonstrated in different studies. In studies focused on cancer development or progression, miR-486-5p acted more like a tumor suppressor. Many studies showed that miR-486-5p could downregulate the proliferation and migration of different cancer cells [30,31,32]. However, in normal cells or tissues, miRNA-486 plays a protective role in different injury models. miR-486-5p could attenuate muscular dystrophy via the PTEN/Akt pathway [33]. miR-486-5p could also reduce hypoxia- or ischemia-induced injury, or cell apoptosis through the suppression of PTEN and activation of Akt [19,34]. Zhang X et al., also indicated that miRNA-486 attenuated hypoxia-induced H9c2 cell injury via JNK and NF-κB downregulation [35]. Our results show that miR-486-5p could induce HPASMC proliferation and migration. However, more data are needed to evaluate which pathway plays an important role in miR-486-5p-induced HPASMC proliferation.

miRNA-486-5p is able to downregulate the TGFβ/Smad2 pathway by directly targeting the 3′-UTR of the Smad2 mRNA [22,24,25]. There are few studies investigating the relationship between miRNA-486-5p and SMAD3. Liu B et al., found that miRNA-486-5p could inhibit Smad3 phosphorylation but not the level of Smad3 expression in lens epithelial cells. The TGFβ/Smad2/3 pathway plays an important role in the promotion of cell proliferation in different injury or disease models [36,37]. The downregulation of Smad2 by miRNA-486-5p also showed effects on the inhibition of cell proliferation and differentiation. In this study, miRNA-486-5p could downregulate the expression of Smad2 and the phosphorylation of Smad3. Those findings are similar to those of a previous study. However, miRNA-486-5p had the opposite effects on HPASMC proliferation and migration. In addition to the possibility that the role of miRNA-486-5p in PAH may be different from that in other injuries or diseases, the compensatory mechanisms for Smad3 downregulation in PAH may also help in explaining these results. Zabini D. et al., indicated that the continued TGFβ activation in PAH could cause the downregulation of Smad3. The loss of Smad3 in PAH promoted cell proliferation and migration via a myocardin-related transcription-factor-independent pathway [29]. A previous study also showed that miRNA-199-5p promoted PAH progression through Smad3 inhibition [38]. Those studies indicated that a loss of Smad3 may promote PAH development. HPASMCs that we isolated from a CHD patient were under chronic hypoxia, which could induce continued TGFβ activation. Smad3 downregulation caused by miR-486-5p may also participate in HPASMC proliferation and accelerate the vascular remodeling in PAH development.

In this study, we investigated the possible role and mechanisms of miR-486-5p in HPASMCs isolated from a CHD patient. We first demonstrated that miR-486-5p overexpression could induce HPASMC proliferation and migration. Although in vivo data are still needed, this study shows the potential roles for and effects of miR-486-5p during PAH development.

## 4. Materials and Methods

### 4.1. HPASMC Isolation and Cell Culture

In vitro studies were performed with human primary HPASMCs that had been isolated from a 6-month 10-day-old boy with a diagnosis of the tetralogy of Fallot. The tissues from main or peripheral pulmonary arteries (PA) were harvested during major cardiac surgery by an experienced surgeon. This study was approved by the Institutional Ethics Committee of National Taiwan University Hospital, Taipei, Taiwan (NTUH201612179RINB), and each participant provided written consent prior to participation. The PA tissues were cut into small pieces and swam with 5 mL of 100 U/mL type I collagenase in Hanks’ Balanced Salt Solution (HBSS) without calcium and magnesium (Gibco, NY, USA). After 30 min of incubation at 37 °C, 10 mL of Medium 231 containing Smooth Muscle Growth Supplement (Gibco, NY, USA) was added into the tube to stop the digestion process, and the samples were pipetted until the solution had turned cloudy. The samples were centrifuged at 250× *g* for 5 min, and the supernatant was carefully removed. The pellets were resuspended in 10 mL of growth medium. After pipetting, the samples were filtered with 70 μm cell strainers to remove the remaining tissue. The isolated cells were incubated with HPASMC selection medium (Medium 231 containing SMGS, 100 U/mL penicillin, and 100 μg/mL streptomycin (Gibco, NY, USA) in a 37 °C incubator with 5% CO_2_) for 2 passages. α-Smooth muscle actin (α-SMA) and smooth muscle myosin heavy chain 11, which are smooth muscle cell markers, were detected using Western blots.

The isolated cells were cultured in Medium 231 supplemented with Smooth Muscle Growth Supplement (SMGS) (Gibco, Grand Island, NY, USA) at 37 °C in a humidified incubator containing 5% CO_2_. Cells from Passages 4 to 8 were used for all the experiments.

### 4.2. Induction of Hypoxia in HPASMCs

Hypoxia was the most commonly used model to mimic pulmonary arterial hypertension in previous studies. Before the experiments, cells were incubated until 80% confluent. For the 6 h hypoxia treatment, PASMCs were cultured in a humidified atmosphere containing 0.5% O_2_ and 5% CO_2_ at 37 °C. The level of O_2_ was controlled using a Ruskinn SCI-tive Hypoxia Workstation (Baker, Stanford, MD, USA).

### 4.3. Levels of miRNAs in HPASMCs Analyzed by Next-Generation Sequencing (NGS)

HPASMCs were cultured in hypoxic conditions (0.5% O_2_) for 6 hours. From these samples, total RNA was extracted by using TRIzol reagent (Thermo Fisher Scientific, IL, USA) according to the manufacturer’s instructions. The miRNA was then sequenced by using Illumina Sequencing-By-Synthesis (SBS) technology. The small RNA library construction, NGS deep sequencing, and bioinformatic analysis were performed by Welgene Biotech. Co., Ltd. (Taipei, ROC). The miRNAs whose expression levels increased or decreased more than twofold compared with the control were chosen as candidates for further analysis.

### 4.4. RNA Extraction and Real-Time Quantitative Polymerase Chain Reaction (qRT-PCR)

Total mRNA was extracted from HPASMCs using a TRIzol reagent (Thermo Fisher Scientific, MA, USA) according to the manufacturer’s instructions. The mRNA expression of ET-1 was detected using a TaqMan system and normalized to that of β-actin. qRT-PCRs for miR-486-5p and U6 (internal control) were performed using an ABI Prism 7500 FAST Sequence Detector (Applied Biosystems, Waltham, MA, USA) according to the protocols of the TaqMan MicroRNA Reverse Transcription Kit and miRNA-specific TaqMan miRNA assays (Life Technologies, Carlsbad, CA, USA). The values of 2^−ΔΔCT^ were calculated and used to represent the expression levels of the mRNAs and miRNAs.

### 4.5. miRNA Mimic Transfection

HPASMCs were incubated in 6 cm dishes until 50–70% confluent; then, the miR-486-5p mimic (5′-UCCUGUACUGAGCUGCCCCGAG-3′) and negative-control mimic (5′-UCACCGGGUGUAAAUCAGCUUG-3′) were transfected into the cells at final concentrations of 1–100 nM using Lipofectamine RNAiMAX (Invitrogen, Waltham, MA, USA) according to the manufacturer’s protocol. After 6 h of transfection, the medium containing Lipofectamine RNAiMAX was replaced with growth medium, and the cells were cultured under a humidified atmosphere of 5% CO_2_ and 95% air at 37 °C. After an additional 48 h, the mRNAs or miRNAs were extracted, and ET-1 and miRNA were analyzed.

### 4.6. Western Blot Analysis

Cells were homogenized in RIPA buffer containing 1% protease inhibitor cocktails (Sigma-Aldrich, St. Louis, MO, USA). The protein levels were quantified using a BCA protein assay kit (Thermo Fisher Scientific, Waltham, MA, USA). Amounts of 40 μg of total cell lysates were loaded and separated with 10% SDS-PAGE, and transferred to polyvinylidene fluoride membranes. The membranes were then blocked with 5% skim milk in TBST buffer at room temperature for 1 h, followed by incubation at 4 °C overnight with the primary antibodies, including an anti-p-Smad2 antibody, anti-p-Smad3 antibody, anti-Smad2 antibody, anti-Smad3 antibody (Cell Signaling, Danvers, MA, USA), and anti-β-actin antibody (Thermo Fisher Scientific, Waltham, MA, USA). Then, the membranes were incubated with horseradish-peroxidase-conjugated IgG secondary antibodies (Thermo Fisher Scientific, Waltham, MA, USA) for 1 h at room temperature. The signals were detected using a chemiluminescence detection kit (Millipore, Burlington, MA, USA). The intensity of the protein was quantified using the ImageJ software.

### 4.7. Cell Proliferation Assays

The cell viability of HPASMCs transfected with the miR-486-5p mimic was evaluated with MTT assay. HPASMCs (1.5 × 10^4^ cells) were seeded in each well of a 24-well plate and cultured for 24, 48, and 72 h after transfection, and then incubated with MTT regent at a final concentration of 0.5 mg/mL for 3 h at 37 °C. Cell viability was calculated from the absorbance (optical density) at 570 nm.

### 4.8. Wound-Healing Assay

The wound-healing assay is a method widely used to mimic the cell migration during wound healing in vivo. Cells were incubated in a 6-well plate until a monolayer had formed. A 200 μL sterile pipette tip was used to create a wound gap. All the cell monolayers were scratched using the same sterile pipette tip, and the wounds were performed in the same direction. After scratching, the cells were gently rinsed twice with PBS to remove the detached cells. Lastly, the cells were incubated for 6 h, and the cell migration rate was determined by calculating the distance that the cells had migrated into the wound area.

### 4.9. ET-1 ELISA

The ET-1 levels in the cells and medium were determined by using an ET-1 colorimetric immunometric ELISA kit (Enzo). The culture medium from transfected HPASMCs was collected immediately and stored at −80 °C until use. Optical density was measured at 450 nm, and the concentrations of ET-1 in the samples were calculated using a standard curve for recombinant ET-1. The final data were normalized to the protein concentration.

### 4.10. Statistical Analysis

The results are expressed as the means ± SEMs. The statistical significance was assessed with Student’s t-test. A *p* value < 0.05 was taken as the threshold level for statistical significance. Apart from NGS data, the experiments in this study were repeated at least three times.

## Figures and Tables

**Figure 1 ijms-23-10400-f001:**
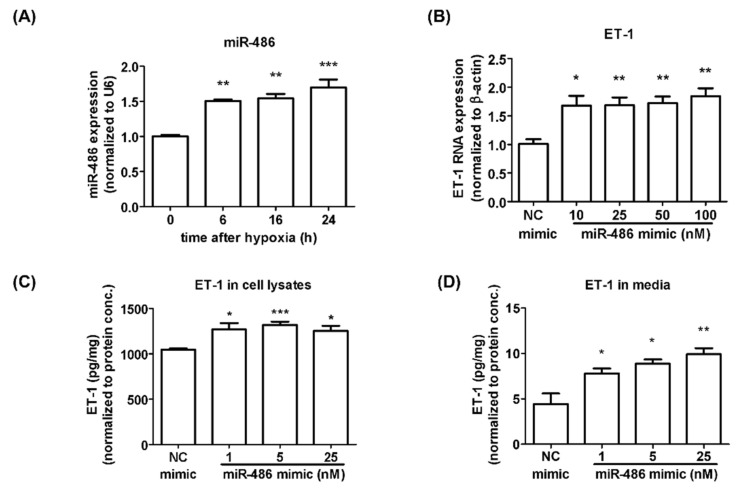
miR-486-5p increased the expression and secretion of ET-1 in HPASMCs. (**A**) After 6, 16, and 24 h of hypoxia treatment, the level of miR-486-5p was evaluated by qRT-PCR. HPASMCs were transfected with a miR-486-5p mimic or NC mimic, and the total RNA, cell lysates, and medium samples were collected after 48 h. (**B**) mRNA expression of ET-1 was determined by qRT-PCR. (**C**) ET-1 protein expression and (**D**) ET-1 secretion were determined by ELISA. Results are expressed as the means ± SEMs (*n* ≥3). * *p* < 0.05, ** *p* < 0.01, and *** *p* < 0.001 compared with the T0 or NC mimic group.

**Figure 2 ijms-23-10400-f002:**
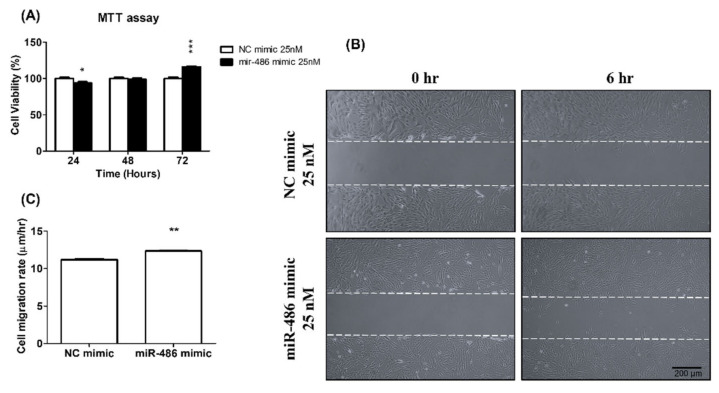
miR-486-5p induced HPAMC proliferation and migration. (**A**) HPASMCs were transfected with the miR-486-5p mimic or NC mimic, and cell viability at 24, 48, and 72 h was determined with an MTT assay (**B**,**C**). The migration ability of HPASMCs was determined with a wound-healing assay. After creating wound gaps, cells were incubated for 6 h, and the cell migration rate was determined by calculating the distance migrated by the cells into the wound area. Scale bar: 200 μm. Results are expressed as the means ± SEMs (*n* ≥3). * *p* < 0.05, ** *p* < 0.01 and *** *p* < 0.001 compared with the NC mimic group.

**Figure 3 ijms-23-10400-f003:**
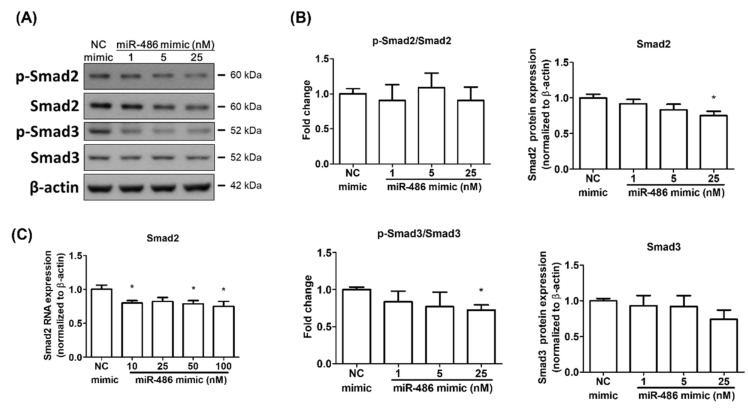
miR-486-5p downregulated Smad2 and Smad3 in HPASMCs. (**A**) HPASMCs were transfected with the miR-486-5p mimic or NC mimic, and the total mRNA and cell lysate were collected after 48 h. The protein expression and phosphorylation of Smad2 and Smad3 were determined with Western blots. (**B**) The intensities of p-Smad2, p-Smad3, Smad2, Smad3, and β-actin were quantified using the ImageJ software. (**C**) The mRNA level of Smad2 was determined by qRT-PCR. Results are expressed as the means ± SEMs (*n* = 4). * *p* < 0.05 compared with the NC mimic group.

**Table 1 ijms-23-10400-t001:** Results of miRNA alterations in HPASMCs under hypoxia.

	Hypoxia/Normoxia		Hypoxia	Normoxia
miRNA	Fold Change	Up/Down	Seq(Normalized)	
hsa-miR-124-3p	−12.38	Down	0.13	1.61
hsa-miR-1306-5p	−2.79	Down	1.01	2.82
hsa-miR-219a-2-3p	−9.77	Down	0.13	1.27
hsa-miR-338-3p	−9.78	Down	0.5	4.89
hsa-miR-4804-5p	−3.25	Down	0.69	2.24
**hsa-miR-486-5p**	**2.8**	**UP**	**4.42**	**1.58**
hsa-miR-6513-3p	3.3	UP	1.32	0.4
hsa-miR-874-3p	2.8	UP	2.27	0.81
hsa-miR-9-3p	−4.52	Down	1.77	8
hsa-miR-9-5p	−5.87	Down	9.97	58.53
hsa-miR-95-3p	−4.72	Down	0.5	2.36

## Data Availability

The data presented in this study are available from the corresponding author upon reasonable request.

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
