# Peer review of "Microrna-486-5P Regulates Human Pulmonary Artery Smooth Muscle Cell Migration via Endothelin-1"

_ijms, 2022, doi:10.3390/ijms231810400_

Round 1

Reviewer 1 Report

It is recommended to reconsider this work if the following comment can be responded, and suggestions can be addressed in revised manuscript.

1)      In Section 2.1, the term “CHD” showed first time without full explanation. Please add some more details to this term to make the readers know better about the source of HPASMCs.

2)      For miRNA profiling, the author only provided significantly changed species. Could the author provide an overview of the sequencing result, such like volcano plot?

3)      In table 1, did the author selected those candidates only by fold change? Did the author perform any statistical analysis for p or FDR values if there were multiple replicates?

4)      In table 1, it is better to highlight miR-486-5p which will be selected for following study.

5)      In figure 1c and 1d, why the author chose lower concentrations of miR-486 for ELISA assay but higher for mRNA expression assay?

6)      In figure 1c and 1d, it was clear that miR-486 increase the protein expression but the current evidence didn’t support it increase the protein secretion. The increased level in medium may be only due to increased protein production rather than promoting its secretion. The author needs to provide more evidence to support that claim.

7)      The whole figure 2 is as same as figure 3. Could the author be patient to go through the manuscript before submitting it?

8)      In figure 3a, did the author normalize the p-Smad2 bands to corresponding Smad2 bands? It looks like to me the downregulation of p-Smad2 was due to decreasing Smad2 rather than a regulation mechanism on its phosphorylation. If yes, please provide more details or evidence to claim “The phosphorylation of smad2 also decreases after miR-486-5p overexpression.”

Reviewer 2 Report

In the present manuscript the authors have made attempt to show documented evidence from their research regarding finding a potential mechanism(s) of the role of Mir-486-5p in PAH. They have modeled the PAH pathophysiology in a hypoxia-induced human PASMC cell culture model and found an increase in Mir-486-5p at 6 h and later time points. Using a Mir-486-5p mimic did show an increase in ET-1 in both the secreted media and within the cells. This finding is also coupled with a reduced protein expression for both Smad2 and 3 and a reduction in phosphorylation in Smad3. The study concluded that miR-486-5p overexpression could induce HPASMCs proliferation and migration, which is a hall mark signature of PAG pathology.

1.      Overall, the manuscript is very poorly drafted and many places do not appear to be prepared for a scientific communication. The authors need to understand minimum requirement of reporting the quality science with proper background and thoughtful evaluation of the experimental data. The whole manuscript struggles with the use of poor English, wrongly used grammar and tense. In the method section, all ‘will be’ needs to be replaced with appropriate past tense. The title of the manuscript seems overly convoluted and does not make any sense to me as to what the study is about to deliver.

2.      The manuscript merely deciphers any new knowledge to our understanding in PAH pathology and possible intervention. Figures 2 and 3 are redundant and simply repetitions. Legend for figure 1B is missing in the figure panel. As indicated by the authors, simultaneous reduction of Smad 2 protein expression and phosphorylation due to the exposure of Mir-486-5p incorporates a confounder to precisely analyze the true phosphorylation level of the target on an unstable total background and I believe that reduction in expression and phosphorylation both happened due to hypoxia-induced cell death. The authors indicated that they have monitored the cell viability at each treatment time point, but I don’t see any data.

3.      How was the WB band signals digitally imaged? More information on catalog number, host and clonality of all the primary antibodies used in the study needs to be provided. Also provide with the approx. mol wt of each target. Show the WB data with an adjacently ran lane of known mol wt standards with at least one up and one below nol wt standards to properly identify the target band of interest with properly annotated name of the target and mol wt.

4.      Wound healing assay: What was the specification of the pipette tip (Size, diameter of the pointed tip etc) that has been used to induce the wound?

5.      Informed consent: Page 8 states, Informed Consent Statement: Not applicable. Whereas the methods section indicated obtaining informed consent from the study participant. This is certainly conflicting. If tissue from only one juvenile donor has been used then how obtaining informed consent from each participant matters?

Round 2

Reviewer 1 Report

The author of manuscript have responded to all comment and revised the manuscript according to my suggestions. It is recommended to publish this work without any further revision.

Author Response

Thank you very much for the comment.

Reviewer 2 Report

The authors made attempt to address the questions and concerns raised by me on the previous version of the manuscript. Though the present version appear much improved, it still has a major discrepancy in the WB related methods and results.

From the WB data provided with the cover letter, it is evident that the total and phspho protein signals for Smad3 have been detected in two different membranes. Similar observation was noted with total and phspho protein signals for Smad2. Similarly beta actin band signals were also detected in separate membranes. Which is not an acceptable protocol for WB band signal intensity normalization? The total and phospho signals for each target must be determined in the same membrane. Since the authors have used all anti rabbit antibodies for the total and phospho Smad2 and Sma3 proteins and have employed ECL-based detection system, stripping the membrane after adding first primary antibody before adding the second could be the only valid protocol assuming the signals must be detected in the same membrane. Beta actin antibody used is anti mouse, but no specific information for both anti-rabbit and anti-mouse secondary HRP conjugated antibodies have been mentioned. Since the results of the WB data is a critical to support the conclusion of the study, with such a poorly controlled and presented data, the current study does not appear to provide any new mechanistic knowledge to PAH progression.

Round 3

Reviewer 2 Report

The responses provided by the authors addressing my concerns and comments appear satisfactory and the revised manuscript has been improved significantly.